# Signal Recovery with Non-Expansive Generative Network Priors

**Jorio Cocola** *
Harvard University
jcocola@seas.harvard.edu

## Abstract

We study compressive sensing with a deep generative network prior. Initial theoretical guarantees for efficient recovery from compressed linear measurements have been developed for signals in the range of a ReLU network with Gaussian weights and logarithmic expansivity: that is when each layer is larger than the previous one by a logarithmic factor. It was later shown that constant expansivity is sufficient for recovery. It has remained open whether the expansivity can be relaxed, allowing for networks with contractive layers (as often the case of real generators). In this work we answer this question, proving that a signal in the range of a Gaussian generative network can be recovered from few linear measurements provided that the width of the layers is proportional to the input layer size (up to log factors). This condition allows the generative network to have contractive layers. Our result is based on showing that Gaussian matrices satisfy a matrix concentration inequality which we term *Range Restricted Weight Distribution Condition* (R2WDC) and which weakens the *Weight Distribution Condition* (WDC) upon which previous theoretical guarantees were based. The WDC has also been used to analyze other signal recovery problems with generative network priors. By replacing the WDC with the R2WDC, we are able to extend previous results for signal recovery with expansive generative network priors to non-expansive ones. We discuss these extensions for phase retrieval, denoising, and spiked matrix recovery.

## 1 Introduction

The compressed sensing problem consists in estimating a signal $y_\star \in \mathbb{R}^n$ from (possibly) noisy linear measurements

$$b = Ay_\star + \eta$$

where $A \in \mathbb{R}^{m \times n}$ is the measurements matrix, $m < n$ and $\eta \in \mathbb{R}^m$ is the noise.

To overcome the ill-posedness of the problem, structural priors on the unknown signal $y_\star$ need to be enforced. One now classical approach assumes that the target signal $y_\star$ is sparse with respect to a given basis. In the last 20 years, efficient reconstruction algorithms have been developed that provably estimate $s$-sparse signals in $\mathbb{R}^n$ from $m = \mathcal{O}(s \log n)$ random measurements [5, 12].

Another approach recently put forward, leverages trained generative networks. These networks are trained, in an unsupervised manner, to generate samples from a target distribution of signals. Assuming $y_\star$ belongs to the same distribution used to train a generative network $G : \mathbb{R}^k \to \mathbb{R}^n$ with $k \ll n$, an estimate of $y_\star$ can be found by searching the input $\hat{x}$ ("latent code") of $G$ that minimizes

---

*Work done while at Northeastern University

36th Conference on Neural Information Processing Systems (NeurIPS 2022).

the reconstruction error

$$\tilde{x} = \arg\min_{x \in \mathbb{R}^x} f_{\text{cs}}(x) := \frac{1}{2}\|b - AG(x)\|_2^2, \tag{1}$$

$$y_\star \approx G(\tilde{x}).$$

As empirically demonstrated in [3], the minimization problem (1) can be solved efficiently by gradient descent methods. Moreover, solving (1) can effectively regularize the solution of the compressed sensing problem, significantly outperforming sparsity-based algorithms in the low measurements regime[3]. Generative network based inversion algorithms have been subsequently developed for a variety of signal recovery problems, demonstrating their potential to outperform inversion algorithms based on non-learned (hand-crafted) priors [16, 31, 30, 20, 33, 28]. For a recent overview see [32].

The optimization problem (1) is in general non-convex and gradient-based methods could get stuck in local minima. To better understand the empirical success of (1), in [18] the authors established theoretical guarantees for the noiseless compressed sensing problem ($\eta = 0$) where $G : \mathbb{R}^k \to \mathbb{R}^n$ is a $d$-layer ReLU network of the form:

$$G(x) = \text{ReLU}(W_d \cdots \text{ReLU}(W_2\text{ReLU}(W_1x))) \tag{2}$$

with $W_i \in \mathbb{R}^{n_i \times n_{i-1}}$, $n_0 = k$, $n_d = n$, and $\text{ReLU}(z) = \max(z, 0)$ is applied entrywise. The authors of [18] used a probabilistic model for the generative network $G$ and measurement matrix $A$. They assumed that each layer $W_i$ has independent Gaussian entries and is *strictly expansive*. Specifically it holds that

$$n_i \geq n_{i-1} \cdot \log n_{i-1} \cdot \text{poly}(d) \qquad \text{for all } i = 1, \ldots, d. \tag{3}$$

Moreover, they considered $A$ to be a Gaussian matrix and $m \geq k \cdot \log n \cdot \text{poly}(d)$. Under this probabilistic model it was shown in [18] that, despite its non-convexity, $f_{\text{cs}}$ has a favorable optimization geometry and no spurious critical points exist apart from $x_\star$ and a negative multiple of it $-\rho_d x_\star$, where $\rho_d$ is a function of the depth $d$ of the network.

The landscape analysis was later extended to recovery guarantees using a gradient based method in [21], under the same probabilistic assumptions of [18]. In particular, [21] has shown that there is an efficient gradient descent method (see Algorithm 1 in Section 3) that given as input $A, G$ and $b$ outputs a latent vector $\tilde{x}$ such that $\|y_\star - G(\tilde{x})\|_2 = O(\|\eta\|_2)$. This result demonstrated that efficient recovery is possible with a number of measurements which is information-theoretic optimal up to $log$-factors in $n$ and polynomials in $d$ ($m = \tilde{\Omega}(k)$).

Generative networks used in practice though, have often contractive layers. For example, the output of the layers near the end of the StyleGAN generators have larger dimension than the generated images [25, 24]. Thus, one major drawback of the theory developed in [18] is constituted by the expansivity condition on the weight matrices (3). Relaxing the condition (3) and accommodating for generative networks with contractive layers was formulated as an open problem in the survey paper[2] [32].

An initial positive result on this problem came from [10]. Using a refined analysis of the concentration of Lipschitz functions, the authors proved that the results of [18, 21] hold true also for weight matrices satisfying $n_i \geq n_{i-1} \cdot \text{poly}(d)$. While not allowing for contractive layers, this condition removed the logarithmic expansivity requirement of (3).

More recently, [22, 23] have studied the denoising and compressive sensing problem with random generative network prior as in [18, 21, 20], and have shown that the expansivity condition can indeed be relaxed. In [23] they have provided an efficient iterative method that given as input $A, b$ and $G$, assuming that up to $log$-factors each layer width satisfies

$$n_i \gtrsim 5^i k, \tag{4}$$

and the number of measurement satisfies

$$m \gtrsim 2^d k, \tag{5}$$

outputs a latent vector $\tilde{x}$ such that for $y_\star = G(x_\star)$ it holds that $\|y_\star - G(\tilde{x})\|_2 = O(2^d\sqrt{\frac{k}{m}}\|\eta\|_2)$ with high probability. Notice that the condition (4) while requiring the width to grow with the depth, can allow for contractive layers $n_i < n_{i-1}$.

---

[2]This open problem was also proposed in the recent talk [11].

## 1.1 Our contributions

It is natural to wonder whether the price to pay to remove the expansivity assumption is indeed the exponential factors in the depth $d$ of the network and the use of less-standard non-gradient based iterative methods, as happens in [22, 23]. In this paper, we answer these questions. Our main result is summarized below and provides guarantees for solving compressed sensing with random generative network priors via a gradient descent method (Algorithm 1 in Section 3).

**Theorem 1.1** (Informal version of Theorem 5.4). *Assume that $A$ has i.i.d. $\mathcal{N}(0, 1/m)$ entries and each $W_i$ has i.i.d. $\mathcal{N}(0, 1/n_i)$ entries. Suppose that $y_\star = G(x_\star)$. Furthermore assume that, up to log-factors,*

1. *$n_i \geq k \cdot poly(d)$;*

2. *$m \geq k \cdot poly(d)$.*

*Suppose that the noise error and the step size $\alpha > 0$ are small enough. Then with high probability, Algorithm 1 with input loss function $f_{cs}$, step size $\alpha$ and number of iterations $T = poly(d)$, outputs an estimate $G(x_T)$ satisfying $\|G(x_T) - y_\star\|_2 = O(\sqrt{\frac{k}{m}}\|\eta\|_2)$.*

Compared to [21] and [10], our result do not require strictly expanding generative networks and allows for contractive layers. Furthermore, we show that the same algorithm proposed in [21] has a denoising effect, leading to a reconstruction of the target signal $y_\star$ of the order $O(\sqrt{\frac{k}{m}}\|\eta\|_2)$ rather than only $O(\|\eta\|_2)$. We show that this holds true even in case of deterministic noise, while [19] discuss only the case of Gaussian noise. Furthermore, the decrease in the reconstruction error with the number of measurements has also been observed for trained generative networks (see for example [3]), and here we give a partial theoretical explanation for this phenomenon.

Compared to the results of [23] we show that it is sufficient for the width of the layers as well as the number of measurements to grow polynomially with the depth rather than exponentially. Similarly, compared to [23], we remove the exponential factor in the depth from the reconstruction error.

The analysis of [18] was based on a deterministic condition on the weight matrices termed *Weight Distribution Condition* (WDC). This condition, together with a deterministic condition on $A$ (see Sec 4 for details), was shown to be sufficient for the absence of spurious local minima in (1) and to be satisfied by expansive Gaussian random generative networks as (2). The WDC was also used in the subsequent [21] to prove convergence of Algorithm 1. Our main technical contribution is to show that the WDC can be replaced by a weaker form of deterministic condition, termed *Range Restricted Weight Distribution Condition* (R2WDC), and still, obtain the absence of spurious local minima and recovery guarantees via Algorithm 1. We will then show that random Gaussian networks satisfying the Assumption 1. of Theorem 1.1 satisfy the R2WDC.

The framework introduced in [18] was used in a number of recent works to analyze other signal recovery problems with generative network priors, from one-bit recovery to blind demodulation [34, 27, 16, 15, 35, 8]. These works considered expansive generative network priors, using the WDC and the results of [18] in their analysis. Replacing the WDC with our R2WDC we can extend the previous results in the literature to more realistic (non-expansive) generative networks. This paper details these extensions for three representative signal recovery problems.

**Theorem 1.2.** *Suppose $G$ is random generative network as in (2), satisfying Assumption 1. of Theorem 1.1. Then Algorithm 1 with appropriate loss functions, step sizes, and number of steps, succeed with high probability for Phase Retrieval, Denoising, and Spiked Matrix Recovery.*

Our result on the denoising problem, implies a similar result on the inversion of a generative network. The problem of inverting a generative neural network has important applications [39, 1, 33], and has been recently analyzed theoretically [26, 22, 2]. Our result shows that a random generative network can be efficiently inverted by gradient descent, even when containing contractive layers. This motivates the empirical use of gradient-based methods for inverting generative networks.

## 1.2 Organization of the paper

This paper is organized as follows. In Section 2 we introduce some notation used in the rest of the paper. In Section 3 we formalize the compressed sensing problem with a generative network prior and describe an algorithm for the recovery. In Section 4 we describe our novel deterministic condition on the weights of the network (R2WDC) and provide theoretical guarantees for solving compressed sensing with a generative network prior satisfying this condition via the algorithm described in Section 3. Then in Section 5 we demonstrate that random non-expansive generative networks satisfy the R2WDC with high probability. The appendix contains the full proof of the results described in the main text. Appendix F contains the extension of the theoretical guarantees for compressed sensing with a generative network prior to other signal recovery problems.

## 2 Preliminaries

We use $I_n$ to denote the $n \times n$ identity matrix. For $j \geq 0$, we define the $j$-th sub-network $G_j : \mathbb{R}^k \to \mathbb{R}^{n_j}$ as $G_j(x) = \mathsf{ReLU}(W_j \cdots \mathsf{ReLU}(W_2\mathsf{ReLU}(W_1 x)))$, with the convention that $G_0(x) = I_k x = x$. For a matrix $W \in \mathbb{R}^{n \times k}$, let $\mathrm{diag}(Wx > 0)$ be the diagonal matrix with $i$-th diagonal element equal to 1 if $(Wx)_i > 0$ and 0 otherwise, and $W_{+,x} = \mathrm{diag}(Wx > 0)W$. We then define $W_{1,+,x} = (W_1)_{+,x} = \mathrm{diag}(W_1 x > 0)W_1$ and

$$W_{j,+,x} = \mathrm{diag}(W_j W_{j-1,+,x} \cdots W_{2,+,x} W_{1,+,x})W_j.$$

Finally, we let $\Lambda_{0,x} = I_k$ and for $j \geq 1$ $\Lambda_{j,x} = \prod_{\ell=1}^{j} W_{\ell,+,x}$ with $\Lambda_x = \Lambda_{d,x} = \prod_{\ell=1}^{d} W_{\ell,+,x}$. Notice in particular that $G_j(x) = \Lambda_{j,x} x$ and $G(x) = \Lambda_x x$.

For $r, s$ nonzero vectors in $\mathbb{R}^\ell$, we define the matrix

$$Q_{r,s} = \frac{\pi - \theta_{r,s}}{2\pi} I_\ell + \frac{\sin \theta_{r,s}}{2\pi} M_{\hat{r} \leftrightarrow \hat{s}} \tag{6}$$

where $\theta_{r,s} = \angle(r, s)$, $\hat{r} = r/\|r\|_2$, $\hat{s} = s/\|s\|_2$, $I_\ell$ is the $\ell \times \ell$ identity matrix and $M_{\hat{r} \leftrightarrow \hat{s}}$ is the matrix that sends $\hat{r} \mapsto \hat{s}$, $\hat{s} \mapsto \hat{r}$, and with kernel $\mathrm{span}(\{r, s\})^\perp$. If $r$ or $s$ are zero, then we let $Q_{r,s} = 0$. The operator $Q_{r,s}$ is used to define the WDC in the next sections, and allows to control how a random ReLU layer distorts its inputs. Specifically, for very $r, s \in \mathbb{R}^\ell$ we have $\mathbb{E}\left[\mathsf{ReLU}(Wr)^T \mathsf{ReLU}(Ws)\right] = r^T Q_{r,s} s$ when $W \in \mathbb{R}^{n \times \ell}$ has i.i.d. $\mathcal{N}(0, 1/n)$.

## 3 Problem statement and recovery algorithm

Consider a generative network $G : \mathbb{R}^k \to \mathbb{R}^n$ as in (2). The compressive sensing problem with a generative network prior can be formulated as follows.

---

**COMPRESSED SENSING WITH A DEEP GENERATIVE PRIOR**

**Let**:    $G : \mathbb{R}^k \to \mathbb{R}^n$ generative network, $A \in \mathbb{R}^{m \times n}$ measurement matrix.

**Let**:    $y_\star = G(x_\star)$ for some unknown $x_\star \in \mathbb{R}^k$.

**Given**:    $G$ and $A$.

**Given**:    Measurements $b = Ay_\star + \eta \in \mathbb{R}^m$ with $m \ll n$ and $\eta \in \mathbb{R}^m$ noise.

**Estimate**:    $y_\star$.

---

To solve the compressed sensing problem with deep generative prior $G$, in [21], the authors propose the gradient descent method described in Algorithm 1 with objective function $f = f_{\mathrm{cs}}$. This algorithm attempts to minimize the objective function $f_{\mathrm{cs}}$ in (1). Because of the ReLU activation function, the loss function $f_{\mathrm{cs}}$ is nonsmooth. Algorithm 1 therefore resorts to the notion of *Clarke subdifferential*. Indeed, being continuous and piecewise smooth, $f_{\mathrm{cs}}$ is differentiable almost everywhere (by Rademacher's theorem) and admits a Clarke subdifferential given by[3]:

$$\partial f_{\mathrm{cs}}(x) = \mathrm{conv}\big\{ \lim_{p \to \infty} \nabla f_{\mathrm{cs}}(x_p) : \ x_p \to x, \ x_p \in \mathrm{dom}(\nabla f_{\mathrm{cs}})\big\}, \tag{7}$$

---

[3]For details see for example [7].

where with conv($\cdot$) we denote the convex hull and with dom($\nabla f$) the subset of $\mathbb{R}^k$ where $f$ is differentiable. The vectors $v_x \in \partial f_{\mathrm{cs}}(x)$ are called the *subgradients* of $f_{\mathrm{cs}}$ at $x$, and at a point $x$ where $f_{\mathrm{cs}}$ is differentiable it holds that $\partial f_{\mathrm{cs}}(x) = \{\nabla f_{\mathrm{cs}}(x)\}$.

---

**Algorithm 1:** SUBGRADIENT DESCENT [21]

---

**Input:** Objective function $f$, initial point $x_0 \in \mathbb{R}^k \setminus \{0\}$ and step size $\alpha$
**Output:** An estimate of the target signal $y_\star = G(x_\star)$ and latent vector $x_\star$
1 **for** $t = 0, 1, \ldots$ **do**
2     **if** $f(-x_t) < f(x_t)$ **then** $\tilde{x}_t \leftarrow -x_t$
3     **else** $\tilde{x}_t \leftarrow x_t$
4     Compute $v_{\tilde{x}_t} \in \partial f(\tilde{x}_t)$
5     $x_{t+1} \leftarrow \tilde{x}_t - \alpha v_{\tilde{x}_t}$
6 **end**
7 **return** $x_t, G(x_t)$

---

Notice that, as described in line 5, Algorithm 1 corresponds to a subgradient descent method with constant step size $\alpha$. Before taking a step in the direction of the subgradient though, the algorithm checks whether the objective function at the current state $x_t$ has a larger value than the value at its negative $-x_t$, and if so it updates the current state with its negative (line 3-4). This negation step allows the algorithm to escape the spurious critical point in a neighborhood of $-\rho_d x_\star$ where $\rho_d \in (0, 1)$, and it is motivated by the landscape analysis of $f_{\mathrm{cs}}$ under the deterministic and probabilistic assumptions that we describe in the coming sections.

## 4 Recovery guarantees under deterministic conditions

The strategy taken in [18] and [21] to analyze the landscape of the minimization problem (1) and the convergence of Algorithm 1, consists in identifying a set of deterministic conditions on the measurements matrix $A$ and the generative network $G$, that ensure that the objective function $f_{\mathrm{cs}}$ is well behaved and Algorithm 1 converges efficiently to an estimate of $x_\star$ and $y_\star$. These conditions are then shown to hold with high probability under probabilistic models for $A$ and $G$. This is akin to the results on compressed sensing with sparsity where, for example, recovery guarantees were developed under the Restricted Isometry Property [4].

The first condition, introduced in [18], is on the measurement matrix $A$ and ensures that $A^T A$ behaves like an isometry over differences of points in the range of a generative network $G$.

**Definition 4.1** (RRIC [18]). A matrix $A \in \mathbb{R}^{m \times n}$ satisfies the *Restricted Isometry Condition* with respect to $G$ with constant $\epsilon$ if for all $x_1, x_2, x_3, x_4 \in \mathbb{R}^k$, it holds that

$$\left| \left\langle \left( A^T A - I_n \right) \left( G(x_1) - G(x_2) \right), G(x_3) - G(x_4) \right\rangle \right| \leq \epsilon \| G(x_1) - G(x_2) \| \| G(x_3) - G(x_4) \|$$

The second deterministic condition introduced in [18] is on the weight matrices of $G$, ensures that they are approximately distributed like a Gaussian, and allows the control of how the layers of the network distort angles.

**Definition 4.2** (WDC [18]). We say that a generative network $G$ as in (2), satisfies the **Weight Distribution Condition** (WDC) with constant $\epsilon > 0$ if for all $i = 1, \ldots, d$, for all $r, s \in \mathbb{R}^{n_{i-1}}$:

$$\| (W_i)_{+,r}^T (W_i)_{+,s} - Q_{r,s} \|_2 \leq \epsilon, \tag{8}$$

Strictly speaking, in [18] the authors define the WDC as a property of a single weight matrix $W$, and then assume that the WDC is satisfied at each layer $W_i$ of $G$. This is equivalent to the definition above and simplifies the introduction of a novel, weaker, condition on the weight matrices, the R2WDC below.

**Definition 4.3** (R2WDC). We say that a generative network $G$ as in (2), satisfies the **Range Restricted Weight Distribution Condition** (R2WDC) with constant $\epsilon > 0$ if for all $i = 1, \ldots, d$, and for all

$x, y, x_1, x_2, x_3, x_4 \in \mathbb{R}^k$ , it holds that

$$
\begin{aligned}
\left|\langle \left((W_i)^T_{+,r}(W_i)_{+,s} - Q_{r,s}\right)u, v\rangle\right| &\leq \epsilon \|u\|\|v\|, \\
\text{where} \quad r &= G_{i-1}(x), \\
s &= G_{i-1}(y), \\
u &= G_{i-1}(x_1) - G_{i-1}(x_2), \\
\text{and} \quad v &= G_{i-1}(x_3) - G_{i-1}(x_4).
\end{aligned}
\tag{9}
$$

Notice that the R2WDC is weaker than the WDC. Indeed, (8) and (9) are equivalent for $i = 1$, but for $i \geq 2$ equation (8) requires $(W_i)^t_{+,r}(W_i)_{+,s}$ to be close to the matrix $Q_{r,s}$ for any vector $r, s \in \mathbb{R}^{n_{i-1}}$ and when acting on any vector $u, v \in \mathbb{R}^{n_{i-1}}$, while equation (9) requires $(W_i)^t_{+,r}(W_i)_{+,s}$ to be close to the matrix $Q_{r,s}$ only for vectors $r, s$ on the range of $G_{i-1}$ and when acting on vectors $u, v \in \mathbb{R}^{n_{i-1}}$ given by the difference of points on the range of $G_{i-1}$. Notice that contrary to (8), defining the R2WDC (9) for layer $i$ requires considering the input/ouput pairs of the layers up to $i - 1$.

Our first technical result provides theoretical guarantees for efficiently estimating a target signal $y_\star$ on the range of a generative network from few linear measurements under the RRIC and the R2WDC .

**Theorem 4.4.** *Suppose $d \geq 2$, and $A$ and $G$ satisfy the RRIC and the R2WDC with constant $\epsilon < K_1/d^{90}$. Assume that $\|\eta\|_2 \leq \frac{K_2\|x_\star\|_2}{d^{42}2^{d/2}}$. Let $\{x_t\}$ be the iterates generated by Algorithm 1 with loss function $f_{\mathrm{cs}}$, initial point $x_0 \in \mathbb{R}^k \setminus \{0\}$ and step size $\alpha = K_3 \frac{2^d}{d^2}$. Then there exists a number of steps $T$ satisfying $T \leq \frac{K_4 f(x_0)2^d}{d^4 \epsilon \|x_\star\|_2^2}$ such that*

$$
\|x_T - x_\star\|_2 \leq K_5 d^9 \|x_\star\|_2 \sqrt{\epsilon} + K_6 d^6 2^{d/2} \omega \|\eta\|_2.
$$

*In addition, for all $t \geq T$, we have*

$$
\|x_{t+1} - x_\star\|_2 \leq C^{t+1-T}\|x_T - x_\star\|_2 + K_7 2^{d/2}\|\eta\|_2,
$$

$$
\|G(x_{t+1}) - y_\star\|_2 \leq \frac{1.2}{2^{d/2}}C^{t+1-T}\|x_T - x_\star\|_2 + 1.2K_7\|\eta\|_2,
$$

*where $C = 1 - \frac{7}{8}\frac{\alpha}{2^d} \in (0,1)$. Here, $K_1, \ldots, K_7$ are universal positive constants.*

**Remark 1.** *The exponential factors $2^d$ appearing in the conditions and theses of the theorem are artifacts of the scaling of the weights of the generative network. For example, the output $G(x)$ of the network scales like $\|x\|_2/2^{d/2}$ and the loss function $f_{\mathrm{cs}}(x)$ as $\|x\|_2^2/2^d$ (see for example Proposition C.1). Hence, for new constants $K_2', K_4'$ the bounds for $\eta$ and $T$ could be equivalently written as $\|\eta\|_2 \leq K_2'\|y_\star\|_2/d^{42}$ and $T \leq K_4' f(x_0)/(d^4\epsilon\|y_\star\|_2^2)$. Choosing the weights of the network to be $\{\sqrt{2}W_i\}_{i\in[d]}$ would remove the $2^d$ factors in the above theorem (and scale the definition of R2WDC).*

This theorem shows that, despite the nonconvexity of the minimization problem (1), if the RRIC and the R2WDC hold with constant $\epsilon$, after $T = O(\epsilon^{-1})$ number of iterations the iterates of the subgradient descent method described in Algorithm 1 enter in a region of local convergence around $x_\star$. Moreover, after a large enough number of steps, $G(x_t)$ gives an estimate of the target signal $y_\star$ up to the noise level $O(\|\eta\|)$.

Theorem 3.1 in [21] shows that Theorem 4.4 holds assuming that the RRIC and the WDC hold. Our first technical contribution is to show that the WDC in Theorem 3.1 of [21], can be relaxed into the R2WDC. Relaxing the WDC into the R2WDC, will enable the relaxing of the expansivity assumption needed to show that the WDC holds for Gaussian generative networks as we demonstrate in Section 5.

We next describe the role of these deterministic conditions in the analysis of the problem (1). The full proof of Theorem 4.4 is given in Appendix C.

### 4.1 Global landscape analysis via the R2WDC

The analysis of [18] and [21] follows the approach recent line of works that analyze the global landscape geometry of non-convex optimization problems arising in statistical and signal recovery problems (see for example [36, 37, 14, 13] and [6] for an overview). The analysis roughly consists of two steps:

i) Showing that $f_{\text{cs}}(x) \approx f_E(x)$ and $\partial f_{\text{cs}}(x) \approx h_x$ uniformly over $x$.

ii) Analyzing the global properties of $f_E(x)$ and $h_x$, and transfer them to $f_{\text{cs}}(x)$ and $h_x$ using the first step.

Here $f_E(x)$ and $h_x$ are continuous functions of $x$, corresponding to the expected value of $f_{\text{cs}}(x)$ and $\partial f_{\text{cs}}(x)$ under Gaussian weights and measurement matrix $A$ (see next section for details) and zero noise. The RRIC and the WDC are used in [18] and [21] to obtain the uniform concentration in the first step, as well as directly proving convexity-like properties of $\partial f_{\text{cs}}(x)$ in the vicinity of $x_\star$.

To illustrate how the WDC and the R2WDC come into play, consider for simplicity the noiseless case $\eta = 0$. Then at a point $x \in \mathbb{R}^k$ where $G$ is differentiable, the gradient of $f_{\text{cs}}$ is given by

$$\nabla f_{\text{cs}}(x) = \Lambda_{d,x}^T A^T (A \Lambda_{d,x} x - A \Lambda_{d,x_\star} x_\star),$$
$$\approx \Lambda_{d,x}^T (\Lambda_{d,x} x - \Lambda_{d,x_\star} x_\star)$$

where $\Lambda_{d,x}$ and $\Lambda_{d,x_\star}$ ar defined in Section 2 and the approximation uses the fact that $A$ satisfies the RRIC with respect to $G$. Then if $G$ satisfies the WDC we have that

$$\nabla f_{\text{cs}}(x) \approx \Lambda_{d,x}^T (\Lambda_{d,x} x - \Lambda_{d,x_\star} x_\star)$$
$$= \Lambda_{d-1,x}^T (W_d)_{+,G_{d-1}(x)}^T (W_d)_{+,G_{d-1}(x)} \Lambda_{d-1,x} x - \Lambda_{d-1,x}^T (W_d)_{+,G_{d-1}(x)}^T (W_d)_{+,G_{d-1}(x_\star)} \Lambda_{d-1,x_\star} x_\star$$
$$= \Lambda_{d-1,x}^T \Big[ Q_{G_{d-1}(x),G_{d-1}(x)} + O(\epsilon) \Big] \Lambda_{d-1,x} x - \Lambda_{d-1,x}^T \Big[ Q_{G_{d-1}(x),G_{d-1}(x_\star)} + O(\epsilon) \Big] \Lambda_{d-1,x_\star} x_\star$$

where the last line used the WDC to control the concentration of $(W_d)_{+,G_{d-1}(x)}^T (W_d)_{+,G_{d-1}(x)}$ and $(W_d)_{+,G_{d-1}(x)}^T (W_d)_{+,G_{d-1}(x_\star)}$. The resulting terms are then controlled again applying the WDC to the the other $d-1$ weights of $G$, so that proceeding by induction over $d$ one obtains

$$\nabla f_{\text{cs}}(x) \approx h_x := \frac{1}{2^d} x - \frac{1}{2^d} \tilde{h}_{x,x_\star}, \tag{10}$$

where $\tilde{h}$ is a deterministic vector field defined in Appendix C.

In Appendix C we show that the R2WDC can be used to control directly the concentration of the terms

$$\Lambda_{d-1,x}^T (W_d)_{+,G_{d-1}(x)}^T (W_d)_{+,G_{d-1}(x)} \Lambda_{d-1,x} x$$

and

$$\Lambda_{d-1,x}^T (W_d)_{+,G_{d-1}(x)}^T (W_d)_{+,G_{d-1}(x_\star)} \Lambda_{d-1,x_\star} x_\star,$$

around their expectation (with respect to $W_d$) obtaining in this way

$$\nabla f_{\text{cs}}(x) \approx \Lambda_{d,x}^T (\Lambda_{d,x} x - \Lambda_{d,x_\star} x_\star)$$
$$= \Lambda_{d-1,x}^T \big[ Q_{G_{d-1}(x),G_{d-1}(x)} \big] \Lambda_{d-1,x} x - \Lambda_{d-1,x}^T \big[ Q_{G_{d-1}(x),G_{d-1}(x_\star)} \big] \Lambda_{d-1,x_\star} x_\star$$
$$+ O(\epsilon \|\Lambda_{d-1,x}\| \|\Lambda_{d-1,x} x\|) + O(\epsilon \|\Lambda_{d-1,x}\| \|\Lambda_{d-1,x_\star} x_\star\|)$$

Then again applying the R2WDC to the other layers of $G$, we can show that (10) still holds. We can then borrow the analysis of $h_x$ from [21] and obtain the same convergence guarantees.

The advantage of using the R2WDC over the original WDC, is that it is satisfied by random generative networks with contractive layers as we demonstrate in the next section.

## 5  Recovery guarantees under probabilistic assumptions

In this section we give probabilistic models for the measurement matrix $A$, generative network $G$, and noise vector $\eta$ that will ensure that the RRIC and the R2WDC are satisfied with high probability and Algorithm 1 efficiently estimate the target signal $y_\star$ up to an error of the order $\tilde{O}(\sqrt{k/m} \|\eta\|)$.

We make the following assumption on the sensing matrix $A \in \mathbb{R}^{m \times n}$.

**Assumptions A.**

**A.1**  *A is independent from $\{W_i\}_{i=1}^d$.*

***A.2*** *A has i.i.d. $\mathcal{N}(0, 1/m)$ entries.*

***A.3*** *There are sufficient number of linear measurements:*

$$m \geq \widehat{C}_\epsilon \cdot k \cdot \log \prod_{j=1}^{d} \frac{e\, n_i}{k}, \tag{11}$$

*where $\widehat{C}_\epsilon$ depends polynomially on $\epsilon^{-1}$.*

Under Assumptions A, the measurement matrix satisfies the RRIC with respect to $G$ with high probability.

**Lemma 5.1** (Consequence of Proposition 6 in [18]). *Let Assumptions A be satisfied. Then A satisfies the RRIC with constat $\epsilon > 0$ with respect to $G$, with probability at least*

$$1 - \hat{\gamma} e^{-\hat{c}\epsilon m}$$

*where $\hat{\gamma}$ and $\hat{c}$ are positive universal constants.*

*Proof.* This result is proved in Proposition 6 in [18] for a number of measurements $m$ satisfying $m \geq C'_\epsilon \cdot k \cdot d \cdot \log \prod_{j=1}^{d} n_j$ where $C'_\epsilon$ depends polynomially on $\epsilon$. To imporove the lower bound on $m$ to (11) it is enough to follow the proof of Proposition 6 in [18] and use the sharper upper bound on the number of affine subspaces in the range of a gnerative network given in Lemma D.1. $\qquad\square$

We then provide a probabilistic model for a generative network $G : \mathbb{R}^k \to \mathbb{R}^n$ as in (2).

**Assumptions B.**

***B.1*** *Each weight matrix $W_i \in \mathbb{R}^{n_i \times n_{i-1}}$ have i.i.d. $\mathcal{N}(0, 1/n_i)$ entries.*

***B.2*** *The first layer satisfies $n_1 \geq \widetilde{C}_\epsilon \cdot k$, and for any $i = 2, \ldots, d$:*

$$n_i \geq \widetilde{C}_\epsilon \cdot k \cdot \log \prod_{j=1}^{i-1} \frac{e\, n_j}{k}, \tag{12}$$

*where $\tilde{C}_\epsilon$ depends polynomially on $\epsilon^{-1}$.*

***B.3*** *The $\{W_j\}_{j=1}^{d}$ are independent.*

Under Assumptions B, the generative network $G$ satisfies the R2WDC .

**Lemma 5.2.** *Fix $0 < \epsilon < 1$. Consider a $d$-layer ReLU network $G$ with weight matrices $\{W_i\}_{i=1}^{d}$. Assume that the $\{W_i\}_{i=1}^{d}$ satisfy Assumptions B. Then $G$ satisfies the R2WDC with constant $\epsilon$ with probability at least*

$$1 - \gamma \left(\frac{en_1}{k}\right)^{2k} e^{-c_\epsilon n_1} - \gamma \sum_{i=2}^{d} \left(\frac{e\, n_i}{k+1}\right)^{4k} e^{-c_\epsilon n_i/2}$$

*where $c_\epsilon$ depends polynomially on $\epsilon^{-1}$ and $\gamma$ is a positive absolute constant.*

We finally conclude with some assumptions on the noise vector $\eta \in \mathbb{R}^m$.

**Assumption C.** *The noise vector $\eta$ is independent from $A$ and the weights $\{W_i\}_{i=1}^{d}$*

The next lemma is used to bound the perturbation of the objective function $f_{\text{cs}}$ and its gradient due to the presence of the noise term $\eta$. These bounds are then used to show that Algorithm 1 leads to a reconstruction of $y_\star$ of the order $O(\sqrt{k/m}\|\eta\|)$.

**Lemma 5.3.** *Suppose $G : \mathbb{R}^k \to \mathbb{R}^n$ satisfies the R2WDC with $\epsilon < 1/(16\pi d^2)^2$ and $d \geq 2$. Let $A \in \mathbb{R}^{m \times n}$ be a matrix with i.i.d. entries $\mathcal{N}(0, 1/m)$ and $\eta \in \mathbb{R}^m$ satisfies Assumption C. Let*

$$\omega := \frac{2}{2^{d/2}} \sqrt{\frac{13}{12}} \sqrt{\frac{k}{m} \log\left(5 \prod_{j=1}^{d} \frac{e\, n_i}{k}\right)}. \tag{13}$$

*Then with probability at least*

$$1 - e^{-\frac{k}{2} \log(5 \prod_{i=1}^{d} \frac{e\, n_i}{k})}$$

*for every $x \in \mathbb{R}^k$ we have that*

$$\langle x, \Lambda_x^T A^T \eta \rangle \leq \omega \|\eta\| \|x\|, \tag{14}$$

*if in addition $G$ is differentiable at $x$ we also have that*

$$\|\Lambda_x^T A^T \eta\| \leq \omega \|\eta\|. \tag{15}$$

Given the previous assumptions, we are now ready to state the main result of this section.

**Theorem 5.4.** *Suppose $d \geq 2$, $\epsilon < K_1/d^{90}$ and $\omega \|\eta\|_2 \leq \frac{K_2 \|x_\star\|_2}{d^{42} 2^{d/2}}$ where $\omega$ is defined in (13).*
*Assume that $A$, $G$ and $\eta$ satisfy Assumptions A, B and C. Then with probability at least*

$$1 - \gamma \Big(\frac{e\, n_1}{k}\Big)^{2k} e^{-c_\epsilon n_1} - \gamma \sum_{i=2}^{d} \Big(\frac{e\, n_i}{k+1}\Big)^{4k} e^{-c_\epsilon n_i/2} - \hat{\gamma} e^{-\hat{c}\epsilon m} - e^{-\frac{k}{2} \log(5 \prod_{i=1}^{d} \frac{e\, n_i}{k})}, \tag{16}$$

*where $\gamma$, $\hat{\gamma}$ and $\hat{c}$ are positive universal constants, the following holds. Let $\{x_t\}$ be the iterates generated by Algorithm 1 with loss function $f_{\mathrm{cs}}$, initial point $x_0 \in \mathbb{R}^k \setminus \{0\}$ and step size $\alpha = K_3 \frac{2^d}{d^2}$. There exists a number of steps $T$ satisfying $T \leq \frac{K_4 f(x_0) 2^d}{d^4 \epsilon \|x_\star\|_2}$ such that*

$$\|x_T - x_\star\|_2 \leq K_5 d^9 \|x_\star\|_2 \sqrt{\epsilon} + K_6 d^6 2^{d/2} \omega \|\eta\|_2.$$

*In addition, for all $t \geq T$, we have*

$$\|x_{t+1} - x_\star\|_2 \leq C^{t+1-T} \|x_T - x_\star\|_2 + K_7 2^{d/2} \omega \|\eta\|_2,$$

$$\|G(x_{t+1}) - y_\star\|_2 \leq \frac{1.2}{2^{d/2}} C^{t+1-T} \|x_T - x_\star\|_2 + 1.2 K_7 \omega \|\eta\|_2,$$

*where $C = 1 - \frac{7}{8} \frac{\alpha}{2^d} \in (0,1)$. Here, $K_1, \ldots, K_7$ are universal positive constants.*

*Proof.* Combining Lemma 5.1, Lemma 5.2 and Theorem 4.4 we obtain Theorem 5.4 with $\omega = 1$ and probability at least

$$1 - \gamma \Big(\frac{e\, n_1}{k}\Big)^{2k} e^{-c_\epsilon n_1} - \gamma \sum_{i=2}^{d} \Big(\frac{e\, n_i}{k+1}\Big)^{4k} e^{-c_\epsilon n_i/2} - \hat{\gamma} e^{-\hat{c}\epsilon m}.$$

Inspecting the proof of Theorem 3.1 in [21], it is easy to see that if Lemma 5.3 holds, then the conclusions of Theorem 5.4 hold with $\omega$ given by (13) and probability at least (16). $\square$

**Remark 2.** *As for Theorem 4.4, the exponential factors $2^d$ are artifacts of the scaling of the weights of the network. Had the entries of $W_i$ been drawn from $\mathcal{N}(0, 2/n_i)$ the $2^d$ factors would not be present.*

**Remark 3.** *Notice that $4k \log(en/(k+1)) \leq 4k \log(n)/\log(2)$ for every $n \geq 2$. Thus if for every $i = 1, \ldots, d$, it holds that*

$$\frac{n_i}{\log(n_i)} \geq \frac{16 \cdot k \cdot c_\epsilon^{-1}}{\log(2)} \tag{17}$$

*the conclusions of the theorem hold with nontrivial probability bounds. In Appendix G we provide an example of a generative network $G$ with contractive layers satisfying both (12) and (17).*

Theorem 5.4 provides guarantees for the efficient recovery of a signal $y_\star$ in the range of a generative network $G$ from few noisy linear measurements, using a nonconvex (sub)gradient descent method. Notice that the intrinsic dimension of the signal $y_\star$ is $k$ (the dimension of the latent space) and the number of measurements required $m$ is proportional to $k$ and information-theoretically optimal up to log factors in the widths of the network and polynomials in the depth. Notice moreover, that up to these factors, the width $n_i$ of each layer of the network is also required to be linear in $k$. This is necessary to ensure that each subnetwork $G_i : \mathbb{R}^k \to \mathbb{R}^{n_i}$ is invertible, and it is weaker than

the assumptions in the previous works that required $n_i$ to be linear in $n_{i-1}$ in order to ensure the invertibility of every single layer.

In Appendix H we empirically verify the predictions of Theorem 5.4, demonstrating how (a practical variant of) Algorithm 1 recover signals $y_\star$ in the range of non-expansive generative networks from undersampled noisy measurements. We show that the recovery is linear in $k/m$ and that in practice the dependence on the depth $d$ of the networks is milder than that predicted by our theory. We leave for future works the establishing of sharper bounds in the depth $d$.

Limitations of the current and previous works on theoretical guarantees for signal recovery with generative networks are the Gaussian assumption on the weights and the absence of biases. Important directions of future research are the inclusion of biases in the generative network and the departure from the Gaussian weights assumptions for more realistic probabilistic models.

## Acknowledgments and Disclosure of Funding

I would like to thank Paul Hand for comments on an earlier version of this manuscript and Babhru Joshi for helpful discussions. This research was partially supported by the NSF award DMS-1848087.

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
