# OpenReview forum: "Signal Recovery with Non-Expansive Generative Network Priors"
_NeurIPS.cc/2022/Conference — NeurIPS 2022 Accept_

### Official Review · Reviewer_s2nj · 2022-07-11

**Rating:** 7
**Confidence:** 5
**Soundness:** 3 good
**Presentation:** 3 good
**Contribution:** 3 good

**Summary:**

This paper considers the problem of inverse problems with generative priors. Prior work has shown that gradient descent recovers the ground truth in poly-time under the assumption of generative models with random weights and sufficient expansion. As real world networks do not satisfy the expansivity assumption, recent work has tried to relax the assumptions. The most relevant prior work [22] allows for contractive layers (i.e., the size of layer $i$ can be smaller than layer $i-1$), but requires the size of layer $i$ to grow exponentially with $i$. This further implies that the number of measurements required for compressed sensing / phase recovery grows exponentially with the depth of the generative model, while traditional results only required linear / quadratic dependence on depth.

This work shows that gradient descent converges in poly time, and only requires the size of each layer and number of measurements to grow polynomially in $d$. The results are novel and significant, and is in my opinion a valuable contribution to the field.

**Questions:**

I have no major questions or concerns-- I am listing my minor concerns as clarifications.

- Line 87 - 90: Can the authors comment on whether the $O(|| \eta ||_2)$ error in [20] appears due to the assumption that $m \approx k \log n$?
Is there something special about the analysis in this paper that allows for the $\sqrt{k/m}$ term which does not appear in [20]?

- It is generally assumed that phase retrieval is a more difficult problem than compressed sensing, for e.g., random generative priors require $kd \log n$ measurements for compressed sensing, but the best known results for phase retrieval require $k d^2 \log n$. I would appreciate a short paragraph on whether a similar phenomenon appears in this paper, as I think it would be an interesting open problem to find the optimal poly dependence on $d$.

- Perhaps for ease of understanding, it is worth clarifying that the benefit of R2WDC is that it takes into account the whole generative model from layer 1 to $i$, as opposed to WDC, which only considers the input / output pair of layer $i$ without considering the previous layers.



**Limitations:**

The limitations are sufficiently stated.

**Strengths And Weaknesses:**

Strengths:
+ This paper considers an open problem in the field of solving inverse problems with generative priors. In my opinion it is sufficiently important and significant.


+ The analysis proposes a new condition, called the Range Restricted Weight Distribution Condition (R2WDC). This condition is seemingly the key to prove that contractive generative models can also be used. To the best of my understanding, the traditional WDC condition required an isometry between each successive layer _for all_ vectors in $\mathbb{R}^{n_i}$, which led to the requirement that $ n_{i+1} \geq c_i n_i \log n_i$. Under the new R2WDC condition, this isometry needs to only hold for vectors in $\mathbb{R}^{n_i}$ that can be generated by the neural network, which allows for the possible contraction between the layers. The idea is intuitive, and I think the paper provides a good explanation for its success.

+ The results are clearly stated and compared to existing results.

Weaknesses:
- In theorem 4.4, the number of steps $T$ is bounded by $2^d$, which is not truly polynomial in the generator parameters. However, this can be forgiven as $d$ is typically on the order of $\log n$.

- This is similar to the above complaint -- theorem 4.4 bounds the norm of the noise as $ || \eta || \leq \frac{||x||}{poly (d) 2^{d/2}}$. This is not a very big problem, but perhaps the claims for the denoising effect of gradient descent (for e.g., in lines 87 - 90) can be toned down considering how the noise norm must be much smaller than the norm of $x$.

- I'm not sure I agree with Remark 1 . How would the R2WDC condition be satisfied if you scaled the weight matrices? As a simpler counter example, if $W \in R^{m \times n}$ is such that $W_{ij} \sim N(0,1/m)$, then $||Wx|| \approx ||x||$. Now, if $W_{ij} \sim N(0,2/m)$, then you get $||Wx|| \approx 2||x||$, and it seems like the requirements in R2WDC would not be satisfied.

---

> ### Author Response · Authors · 2022-08-02
> **Response to Reviewer s2nj**
>
> We really appreciate your positive feedback and the careful reading of our paper. Below are our answers to the questions and comments raised.
>
>
> "On **Weakness**"
>
> - Regarding the dependence from $2^d$ in the number of iterations $T$ and the size of the noise $\| \| \eta \| \|$, notice that because of the ReLU layers the norm of the output $G(x_\star)$ of the network scale like $\| \|x_\star\| \|/2^{d/2}$, and similarly the loss function $f_{CS}(x_0)$ as $\| \|x_0\| \|^2/2^d$ (see Proposition C.1). Therefore up to change of constants, these bounds for can be written as $T\leq f(x_0)/(d^4 y_\star \epsilon)$ and as $ \| \|\eta \| \|\leq \| \|y_\star\| \|/d^{42}$. We have modified Remark 1 clarifying this point.
>
> ${}$
>
> - Regarding the R2WDC scaling. Notice that with the scaling of the paper one has $\| \| \text{ReLU}(W x)\| \approx \|\| x \| \|/\sqrt{2}$. With the $\mathcal{N}(0, 2/m)$ scaling one would obtain  $\| \| \text{ReLU}(W x)\| \approx \|\| x \| \| $. The R2WDC then would still hold (modulo multiplying the operator $Q_{r,s}$ by a factor of 2).
>
>  "On **Questions**"
>
> - The term $\sqrt{k/m}  \, O(\|  \|  \eta \|  \|) $ (as opposed to $O(\|  \|  \eta \|  \|) $) appearing in the reconstruction bound follows from a more careful analysis of the perturbation of the gradient due to $ \eta $. The perturbation is given by $\Lambda_x^T A^T \eta$. In [20] it is shown that  $\Lambda_x^T A^T$ is $O(1)$. Here we notice that $A^T \eta$ is gaussian of dimension $m$ and $\Lambda_x^T$ "projects it" on subspaces of dimension $k$. This leads to a "denoising effect" of the order $\sqrt{k/m}$.
>
> ${}$
>
> - We agree that understanding the optimal poly dependence on the depth $d$ is an interesting and under-developed area of research. However, notice that both [R20] for compressed sensing and  [R17] for phase retrieval establish a sample complexity of the order $m \geq C_\epsilon d k \log(n_1..n_d)$. So both have a linear explicit dependence on $d$. On the other hand, the factors $C_\epsilon$ in the two papers differ and have implicit dependence on $d$ through $\epsilon$, so really the sample complexity can be thought of as $m \geq poly(d) k \log(n_1..n_d)$. We are not aware of any papers establishing the sharper linear dependence on $d$ for compressed sensing and phase retrieval.
> Regarding this paper, we use a more refined counting of the number of subspaces containing the range of $G$ and show that the sample complexities are of the order $m \geq poly(d) k \log(n_1/k..n_d/k)$.
>
> ${}$
>
> - Thank you for the suggestion! We have expanded the paragraph after the definition of the R2WDC (lines 195-195) commenting on this point.

---

### Official Review · Reviewer_NCFH · 2022-07-12

**Rating:** 7
**Confidence:** 2
**Soundness:** 3 good
**Presentation:** 3 good
**Contribution:** 3 good

**Summary:**

This paper presents a theoretical analysis for signal recovery with non-expansive generative networks. The main results suggest that given a random Gaussian generator, any signal in its range can be reconstructed from Gaussian measurements as long as the number of measurements and the width of all layers are proportional to the size of input layer. This result improves upon the earlier analyses that require width of layers to expand.

**Questions:**

I did not check the derivations for accuracy, so I do not have any specific question at this point.

**Strengths And Weaknesses:**

Strengths.
- The paper presents a new analysis for the general signal recovery using generative priors.
- The main contribution is in relaxing the conditions on the width of the network layers.

Weaknesses.
- This analysis makes a strong assumption that the network has Gaussian weights. This is quite far from real settings where the network is learned from some data. Authors acknowledge that as a limitation. It will be good for the community to start analyzing this real problem.

---

### Official Review · Reviewer_xC1X · 2022-07-12

**Rating:** 5
**Confidence:** 2
**Soundness:** 3 good
**Presentation:** 2 fair
**Contribution:** 2 fair

**Summary:**

This paper relaxes an assumption in previous theoretical work on signal recovery with generative networks. Namely, previous work required that the generative network have constant-width layers (and before that, logarithmically expanding layers). This work shows that the generative model may preserve the signal recovery guarantees, while having layers of contracting width.

**Questions:**

Please answer the motivation question above.

**Limitations:**

No.

**Strengths And Weaknesses:**

Strengths:
- Solid theoretical result on a relaxed assumption for generative model-based signal recovery.

Weaknesses:
- Motivation is lacking. The paper asserts that relaxing the assumption in question is important, as contractive layers are "often the case in real generators". However, this is stated without citation, and I am not sure it is true. Most GANs and VAEs involve constant or growing width of layers in the generative network. The only provided support for relaxing this assumption is a citation of [1]. However, I could not find this statement anywhere in the cited work. The authors did state that relaxing the logarithmically expansive width assumption was important, but that has apparently already been shown by [2, 3].

Overall, it is unclear to me how important the theoretical improvement is, as this is not my main area of expertise, and the paper does not properly motivate the contribution. However, I believe that this work provides a straightforward contribution to theory in this field.

[1] Hand, P., & Voroninski, V. (2018, July). Global guarantees for enforcing deep generative priors by empirical risk. In Conference On Learning Theory (pp. 970-978). PMLR.
[2] Daskalakis, C., Rohatgi, D., & Zampetakis, E. (2020). Constant-expansion suffices for compressed sensing with generative priors. Advances in Neural Information Processing Systems, 33, 13917-13926.
[3] Joshi, B., Li, X., Plan, Y., & Yilmaz, O. (2021, October). PLUGIn-CS: A simple algorithm for compressive sensing with generative prior. In NeurIPS 2021 Workshop on Deep Learning and Inverse Problems.

---

> ### Author Response · Authors · 2022-08-02
> **Response to Reviewer xC1X**
>
> We thank Reviewer xC1X  for the positive feedback on the theoretical results of this paper.
>
> Regarding our contribution and motivations. In summary, previous theoretical works for efficient recovery with (random) generative network priors were only given for expansive networks [R20] (while generative networks used in practice have contractive layers) or for non-gradient descent methods [R22] (many signal recovery methods based on generative networks are based on gradient descent methods). The motivations and contributions of this paper are to fill the gap between empirical results and the (though stylized) theory.
>
> Below are more details on our contributions and motivations.
>
>
> - We notice that modern state-of-the-art generative networks have layers near the outputs that are often larger than the output itself. These networks are therefore non-expansive. For example, in the StyleGAN2 architecture trained on 3 × 256 × 256 images, the output of the second to last layer has dimensions 64 × 256 × 256.  We have added a comment on the expansivity of commonly used generative networks on lines 57-59.
>
> ${}$
>
> - Regarding the expansivity assumption, we notice that [1] requires strict logarithmic expansivity, while [2] requires strict constant expansivity. In this paper, we show that *no strict expansivity* is required.
>
> ${}$
>
> - Finally, while [R22/3] was also able to remove the assumption on the expansivity of the networks this was at the expense of exponential dependence on the depth and the use of a non-standard iterative method. Algorithm 1 of this paper instead is a gradient descent method, closer in spirit to the ones often used in practice and does not suffer from exponential dependence on $d$.
> We have added a remark on these contributions on lines 77-81
>
> ${}$
> ${}$
>
> [1] Hand, P., & Voroninski, V. (2018, July). Global guarantees for enforcing deep generative priors by empirical risk. In Conference On Learning Theory (pp. 970-978). PMLR.
>
> [2] Daskalakis, C., Rohatgi, D., & Zampetakis, E. (2020). Constant-expansion suffices for compressed sensing with generative priors. Advances in Neural Information Processing Systems, 33, 13917-13926.
>
> [3] Joshi, B., Li, X., Plan, Y., & Yilmaz, O. (2021, October). PLUGIn-CS: A simple algorithm for compressive sensing with generative prior. In NeurIPS 2021 Workshop on Deep Learning and Inverse Problems.

---

### Official Review · Reviewer_V26y · 2022-07-25

**Rating:** 5
**Confidence:** 4
**Soundness:** 3 good
**Presentation:** 2 fair
**Contribution:** 3 good

**Summary:**

The paper extends the previous theoretical studies around convergence of subgradient descent (Algorithm 1 from [20]) for ReLU generative priors to cases where the generative model is not necessarily expansive. The result relies on the weights being drawn from i.i.d. Gaussian matrices and assumes network widths and measurement numbers are proportional to input dimension (See Theorem 1.1, conditions 1 and 2). The results extend to other inverse problems like phase retrieval, denoising and spiked matrix recovery.
Theorem 5.4 provides the result for random Gaussian matrices and weights, and Theorem 4.4 provides a more general result for any matrices satisfying RRIC and R2WC. Lemma 5.1 is well known in the literature. Lemma 5.2 and 5.3 are proven in the appendix and together with Theorem 4.4 imply Theorem 5.4.
The extensions to phase retrieval, denoising and spiked matrix recovery are given in the supplementary materials, although the proofs are not explicitly given.


**Questions:**


* [21] seems to be the extended NeurIPS version of workshop paper [22]. It is probably better to use this version instead of [22].
* I suggestion expanding on the denoising effect of the algorithm 1 as mentioned in page 3, line 89. This is an interesting consequence.
* A relevant question, from practical perspective, is to see if one can verify R2WDC for a pre-trained network (not a random one). It is known that verifying RIP property of a matrix is NP-hard. It would be interesting if the authors can comment on it.
* Gradient descent-based methods used in context of trained generative priors suffer from lack of convergence and usually require occasional restarts. This is in contrast with the current claims of optimality in the paper. This merits some comments from the paper. Which assumption in the theoretical framework is likely to be violated for this to happen? Gaussian assumption is a good candidate, but it is just an instance of distributions satisfying R2WDC. Is it related to details of Algorithm 1, for instance the check $f(-x_t)<f(x_t)$? Some comments on this can be helpful for the paper.
* In page 2, the authors mention that the method of [20] is information theoretically optimal given $m=\tilde{\Omega}(k)$. As far as I can see [20] does not claim information theoretic optimality. Why is this the case? Is there any work providing lower bound on the sample complexity for random generative models (for example like Gelfand width analysis in compressed sensing)? A similar claim is made in the final part of the paper.
* Please add a few sentences to the paper on how the subgradient can be computed in practice. Of course, for smooth generative models, this is not an issue, but for ReLU networks, the question is whether the gradient descent as implemented in backprop is a good proxy.
* I suggest rephrasing the mention of Rademacher’s theorem in line 149, page 4. The implication of Rademacher’s theorem, roughly, is that Clarke’s subdifferential is well behaved since the $\text{dom}(\nabla f)$ has full measure.  With current phrasing, this point is not clear.
* I suggest adding the intuition behind the matrix $Q_{r,s}$ to the main paper (for example by mentioning the connection with measuring angle distortion by $x\to W_{+,x}$).
* The $\epsilon$ in Theorem 4.4 is $O(1/d^{90})$. This term would dominate the rest of terms in the sample complexity condition (11) (see Assumption A.3). Basically, $m\geq \hat{C}_\epsilon$ implies at least $m\geq d^{90}$. This is a poor dependency on $d$, although it is present in the related works too.
* In Remark 1 and 2, it is suggested that the factors $2^d$ can be removed with the entries of $W_i$ drawn from $\mathcal{N}(0,2/n_i)$. Would not this violate the requirements for R2WDC condition?
* Apart from $\hat{C}_\epsilon$, the condition A.3 of Assumptions A has implicit linear dependence on $d$. Consider a non-expansive network with $n_i=n$ for all $i$. Then the sample complexity lower bound includes $dm$. Similar arguments can be made of assumption B.2 and for $n_i$, namely $n_i\geq k.i \log (ne/k)$. The authors should comment on this.
* Typo: in two places in the paper, RRWDC is used instead of R2WDC.


**Limitations:**

Although the paper is theoretically interesting and a step forward, I feel that the authors can do a better job in communicating the idea, extracting useful guidelines and presenting the limitations.

**Strengths And Weaknesses:**

**Strength**

* The paper improves the analysis of [22] (also [21]) and gets much better dependence on the number of layers $d$ and removes the dependence of width of layer $i$ on the layer index $i$.
* R2WDC is weaker that WDC condition, and nonetheless provides better results.
* The proof builds on many previous works, for example [18], [20], [22], and, based on my rapid read, is sound and well presented.
* The contribution of the paper, namely relaxing the constraints on the network further and introducing R2WDC, is a good step forward in this analysis.

**Weakness**

* Some important limitations of the paper are not mentioned clearly by the authors, and some of the statements are not fully precise (see some comments below – for example, it seems to me that the sample complexity has drastic dependency on $d$; also on information theoretic optimality.).
* The authors do not extract relevant guidelines from their theory. Practical generative models are trained from data. The paper considers generative models with random Gaussian weight, which is fine to derive the theoretical analysis. Even if the results are not directly applicable to practical networks, it is important to extract theoretical insights from the developed theory (like the authors’ comment on denoising effect of Algorithm 1 and generative priors). This is missing from the paper. See also my comments on gradient descent issues for trained generative priors.
* Having numerical results, for instance similar to those in [20], can always help communicating the paper’s contribution better.
* The paper could have been organized better by removing some discussions to the supplementary materials and presenting first the core idea behind the proof (similar to [20]).

---

> ### Author Response · Authors · 2022-08-02
> **Response to Reviewer V26y**
>
> Thank you for carefully reading our paper and the many interesting questions. Below our comments.
>
> - Regarding citation [21]. Thank you for pointing this out. We have moved this citation to the main body of the paper (line 67-68).
>
> - Regarding the complexity of certifying the R2WDC. Great question! Indeed establishing that the R2WDC is satisfied could also be in general hard. We leave this problem for future work and hope that, as for the RIP, even if it is demonstrated that certifying the R2WDC is hard, it will prove to be useful to better understand the performance of signal recovery methods based on generative networks.
>
> - Regarding the theoretical assumptions on the weight matrices. The fundamental assumptions on the weights of G used in the analysis are the symmetry of the Gaussian distribution and its strong concentration properties. These properties lead to non-convex but well behaved minimization problems like (1) whose geometry suggests checking the condition $f(x_t) < f(-x_t)$ in Algorithm 1.
>
>
> - Regarding rate-optimality. To clarify, in [20] the authors claim that the $m = \widetilde{\Omega}(k)$ is rate-optimal or information-theoretically optimal with respect to $k$ (up to log factors in $n$ and polynomials in $d$). This is indeed optimal, as the following reasoning roughly shows. Consider an oracle that would give the subspace in the range of $G$ in which the target signal $y_\star$ lies. With this information, the problem becomes a simple linear problem over a fixed subspace and would require a number of measurements exactly $k$. If one had then less than $k$ number of measurements, then recovering $y_\star$ exactly would be impossible as fixing all the degrees of freedom would be impossible.
>
> - Regarding Radamacher's theorem. Thank you for the suggestion, we have rephrased the discussion around the Clarke subdifferential (see line 149-150).
>
> - Regarding the R2WDC with the $\mathcal{N}(0, 2/n_i)$ entries. Yes, strictly speaking with this scaling the definition of the R2WDC would need a “rescaling”, notice indeed that with this scaling the expected value of $W_{+,s}^T W_{+,r}$ is $2 Q_{r,s}$.
>
> - Regarding the polynomial scaling with respect to the depth $d$ of the network. As in the previous literature, these have not been optimized and likely to be sub-optimal. As we mention at the end of the paper, we leave establishing sharper bounds for future works.
>
> - Thank you for noticing those two typos in the appendices.
>
> - Regarding the interpretation of $Q_{r,s}$, the practical implementation of Algorithm 1 we refer the reader to the original paper where these were defined. Similarly, since the sketch of the proof of convergence of Algorithm 1 was already given in [20], we prefer to focus on the novel contributions and ideas  of this paper (e.g. discussion in Section 4.1).

---

> > ### Comment · Reviewer_V26y · 2022-08-08
> > **Response to the authors**
> >
> > I would like to thank the authors for their comments ans answers!.
> >
> > * Regarding the argument around information-theoretic optimality: the provided argument works for linear models (with oracle known subspace and solving linear systems etc.). I am not sure if this holds for nonlinear networks and the limit can be worse then (this can be good for the paper: say hypothetically, the IT bound would  depend on $k$ and be logarithmic in $n_i$- in this case, the obtained bound of the paper could be actually claimed to have IT optimal dependencies there). I think the argument for information theoretic limits should way mathematically more rigorous. As far as I can see, I cannot find a mathematical proof of information-theoretic limits. Of course, I would be more than happy if the author could point me to this argument.
> > * I still prefer that the authors to discuss the intuition behind $Q_{r,s}$  in the main paper. Let me ask a question more directly: what is the intuition behind $Q_{r,s}$? Can you please comment here? Note that this can probably help coming up with new practices say adding a regularization term during training.
> >
> > Of course the paper relaxes some of the existing issues of the current theory (expansiveness), it still has many limitations and undesirable dependencies as mentioned by the other reviewers. This is of course natural for theoretical developments, and I would still favor the acceptance, if the author can clarify further what the current theory, despite its limitation, can tell us about the practice of generative priors.

---

> > > ### Author Response · Authors · 2022-08-08
> > > **Response to Reviewer V26y (Lower bounds, Q_{rs}, Numerical Experiments, and more..)**
> > >
> > > Dear reviewer,
> > >
> > > below are the answer to your questions
> > >
> > > - Previous theoretical results proving $m = \widetilde{\Omega}(k)$ lower-bounds were given in [Ra] and [Rb]. Notice that studying (sharp) information-theoretic limits is beyond the scope of this paper, which instead is devoted to analyzing the performance of practical gradient descent methods for solving signal recovery problems with generative network priors.
> > >
> > > - $Q_{r,s}$ is the expected value of $W_{+,r}^T W_{+,s}$ under Gaussian weights distribution. It allows to control how a ReLU layer distorts angles. In particular, $r^T Q_{r,s} s$ is the expected value of $ReLU(W r)^T ReLU(W s)$. We have added a remark on this in Section 2 at lines 142 - 143.
> > >
> > > Regarding the dependencies on the network widths and depth in the proved bounds. One of the objectives of this paper was to prove sample complexities linear in the latent dimension k of the signal and polynomial in the depth d.  We agree that the dependencies on the network widths and depth are undesirable, and we leave getting sharper bounds for future works.
> > > ${}$
> > >
> > > Regarding extracting relevant guidelines from our theory. The main motivation of this paper was to understand the empirical observation that gradient-descent methods can be successfully used to solve compressive sensing (and other signal recovery problems) with generative network priors, _despite_ the non-convexity of the loss functions minimized. Notice that even if in practice one needs a few random restarts of gradient descent to obtain good results, it is still surprising that such highly non-convex functions can be minimized (potentially it is NP-Hard). So how does this informs the use of generative networks? Our theory demonstrates that the loss landscape of minimization problems such as (1), while non-convex can still be well-behaved as long as the weights have well-behaved distributions and the contractive layers and number of measurements are not too small compared to the latent dimension of the generative network. Inspired by our results, one could study methods for regularizing the distribution of the weights of a generative network, to make easier the minimization of (1).
> > >
> > > ------
> > >
> > > ### **Other remarks**:
> > >
> > > #### _Numerical Experiments_
> > >
> > > Based on some of your previous comments we have added a section in the appendix (Appendix H) with synthetic experiments similar to those in [20]. These experiments validate our theoretical findings and show that they hold in a wider parameters range (in particular with a milder dependency on the depth d).
> > >
> > > In Appendix H we also discuss how to compute the subgradients of the loss function, and give a practical algorithm that instead uses the "derivatives" as computed by commonly used deep learning libraries.
> > >
> > > #### _Denoising effect_
> > >
> > > As initially suggested we have expanded on the denoising effect of Algorithm 1 (line 92 - 95). We demonstrate it in the synthetic experiments (Appendix H). We moreover notice that this denoising effect has already been empirically observed in previous works for trained generative networks (e.g. [3]). Hence, our theoretical results provide theoretical insights into this phenomenon.
> > >
> > > ------
> > > References:
> > >
> > >  [Ra] A. Kamath, E. Price, and S. Karmalkar, “On the power of compressed
> > > sensing with generative models”, in International Conference on Machine Learning, 2020.
> > >
> > >  [Rb] Z. Liu and J. Scarlett, “Information-theoretic lower bounds for compressive sensing with generative models”, in IEEE Journal on Selected Areas in Information Theory, vol. 1, no. 1, pp. 292–303, 2020.

---

### Meta-Review · Area_Chair_3wDz · 2022-08-26

**Recommendation:** Accept
**Confidence:** Certain

**Metareview:**

This paper focuses on theoretically studying signal reconstruction with non-expansive generative networks. In short the authors show that with a random Gaussian generator, any signal in its range can be reconstructed from Gaussian measurements. This holds as long as the number of measurements and the width of all layers are proportional to the size of input layer. Compared to prior work this paper removes the requirement of expansion of the layers. Most reviewers thought the paper was interesting and thought the improved theoretical analysis was nice. The reviewers also raised a variety of technical concerns most of which was addressed during the rebuttal. I concur with the reviewers and think is a nice contribution despite some flaws and am recommending acceptance. I urge the authors to follow the details comments of the reviewers to improve their manuscript for the camera ready version of the paper.

**Award:**

No

---

### Decision · Program_Chairs · 2022-09-14

Accept